# Regulatory Role of a Hydrophobic Core in the FliG C-Terminal Domain in the Rotary Direction of a Flagellar Motor

**DOI:** 10.3390/biom15020212

**Published:** 2025-02-01

**Authors:** Tatsuro Nishikino, Akihiro Hatano, Seiji Kojima, Michio Homma

**Affiliations:** 1Department of Life Science and Applied Chemistry, Nagoya Institute of Technology, Gokiso, Showa-ku, Nagoya 466-8555, Japan; 2OptoBioTechnology Research Center, Nagoya Institute of Technology, Gokiso, Showa-ku, Nagoya 466-8555, Japan; 3Department of Biological Science, School of Science, Nagoya University, Furo-cho, Chikusa-ku, Nagoya 464-8602, Japan; 4Department of Biological Science, Graduate School of Science, Nagoya University, Furo-cho, Chikusa-ku, Nagoya 464-8602, Japan; z47616a@cc.nagoya-u.ac.jp; 5Department of Physics, Graduate School of Science, Nagoya University, Furo-cho, Chikusa-ku, Nagoya 464-8602, Japan; g44416a@cc.nagoya-u.ac.jp; 6Department of Biomolecular Engineering, Graduate School of Engineering, Nagoya University, Furo-cho, Chikusa-ku, Nagoya 464-8602, Japan

**Keywords:** flagellar motor, switching protein, FliG, rotational direction, MFXF motif, chemotaxis, *Vibrio*

## Abstract

A flagellar motor can rotate either counterclockwise (CCW) or clockwise (CW), and rotational switching is triggered by conformational changes in FliG, although the molecular mechanism is still unknown. Here, we found that *cheY* deletion, which locks motor rotation in the CCW direction, restored the motility abolished by the *fliG* L259Q mutation. We found that the CCW-biased *fliG* G214S mutation also restored the swimming of the L259Q mutant, but the CW-biased *fliG* G215A mutation did not. Since the L259 residue participates in forming the FliG hydrophobic core at its C-terminal domain, mutations were introduced into residues structurally closer to L259, and their motility was examined. Two mutants, D251R and L329Q, exhibited CW-biased rotation. Our results suggest that mutations in the hydrophobic core of FliG_C_ collapse its conformational switching and/or stator interaction; however, the CCW state of the rotor enables rotation even with this disruption.

## 1. Introduction

The bacterial flagellum, a key motility machine, is utilized for swimming in liquids or swarming on solid surfaces, enabling movement toward favorable conditions for cell survival and growth [1]. The flagellum comprises more than 20 proteins and is divided into three parts: the basal body, which functions as a rotary motor; the filament, which serves as a helical propeller for generating propulsion power for the cell to move; and the hook, which connects the basal body to the filament and transmits motor torque to the filament. The flagellar motor is one of the models for the energy conversion system utilizing protein–protein interaction, and cell flagellar motility is involved in infection and pathogenicity in pathogenic bacteria; thus, examining this rotation mechanism leads to a better understanding of these bacteria.

Multiple stator units surround the basal body and function as ion channels that couple ion flux through the stator channel with torque generation; each unit converts the electrochemical gradients of specific ions (protons or sodium ions) across the membrane. The stator comprises two membrane proteins, motility proteins A and B [1,2,3]. The A and B subunits in the proton-driven stators from *Escherichia coli* and *Salmonella*, as well as in the sodium-driven stator from *Vibrio alginolyticus*, are called MotA and MotB or PomA and PomB, respectively. The A subunit is a four-transmembrane protein, and its cytoplasmic region between the second and third transmembrane segments interacts with FliG in the rotor to transmit motor torque [1,4,5,6,7,8]. The B subunit has a single transmembrane (TM) segment and a peptidoglycan-binding (PGB) domain, which is connected by a plug helix and an intrinsically disordered linker. Stator incorporation into the motor triggers a conformational change in the linker, allowing the PGB domain to anchor to the cell wall and enabling channel unplugging to conduct ions for torque generation [1,9,10]. During this stator activation step, FliL, a single transmembrane protein, helps the stator to anchor around the rotary motor, presumably by interacting with the B subunit [11,12,13]. Recent structural analyses of stator complexes have revealed that a single stator unit is a hetero-heptamer complex: two B subunit transmembrane segments are surrounded by a ring structure composed of five A subunits [14,15,16,17]. The structure implies the clockwise rotation of the pentameric A subunit ring against the B subunit axis that couples with the ion flux through the stator channel [14,15,16,17,18].

The rotor part of the motor comprises several ring complexes: the membrane-embedded MS-ring, which is mounted on the cytoplasmic C-ring, and the LP-ring, which penetrates through the peptidoglycan layer and the outer membrane. Marine *Vibrios* have accessory components on the polar flagellar basal body, such as the T-ring and H-ring [1], to adapt to the high-speed rotation of the motor. The flagellar motor of *Vibrio* can rotate counterclockwise (CCW) and clockwise (CW). The cell swims smoothly and exhibits tumble motion to switch directions when the motor rotates in the CCW and CW directions. Among these ring complexes, the C-ring comprises three proteins (Figure 1A) [19]: FliG, FliM, and FliN. These play a role in determining the rotational direction of the motor, which is triggered by chemotactic signals. When cells receive environmental signals, the two-component system transmits the signals to the motor in the form of phosphorylated CheY (CheY-P), one of the Che factors. CheY-P binding to FliM and FliN induces the CW rotation of the motor owing to the conformational rearrangement of the C-ring [18,20,21,22,23,24,25,26]. CheY-P dephosphorylation induces CheY dissociation from FliM and FliN, followed by motor rotation in the CCW direction.

During rotational switching, FliG undergoes conformational changes that alter the rotor–stator interaction surfaces. According to the full-length structure of FliG from *Aquifex aeolicus* [27], FliG can be divided into three domains: the N-terminal (FliG_N_), middle (FliG_M_), and C-terminal (FliG_C_) (Figure 1B). The hydrophobic armadillo repeat motifs (ARMs) found in each domain—namely, ARM_N_, ARM_M_, and ARM_C_, respectively—participate in intramolecular and/or intermolecular interactions and contribute to the formation of the domain-swap oligomerization of FliG molecules in the C-ring structure [28,29]. FliG_N_ interacts with the C-terminal cytoplasmic tail of FliF [1,30,31,32], an MS-ring component [33,34,35,36,37,38]. The EHPQR motif in FliG_M_ interacts with FliM, and this binding is affected by receiving/releasing signals on FliM, a way of binding CheY-P or releasing CheY [1,39]. FliG_M_ and FliG_C_ are connected via Gly-Gly flexible linkers (G214 and G215 in marine *Vibrio*). The conserved MFXF motif in FliG_C_ separates FliG_C_ from FliG_CN_ and FliG_CC_. These residues and motifs are important for determining the rotational direction. FliG_CC_ contains conserved charged residues that electrostatically interact with the stator; they are localized on a helix that faces the stator, referred to as the “torque helix”. Such electrostatic interactions are critical for motor rotation, and thus, the orientation of the charged residues on the torque helix of FliG_CC_ is important for determining the rotational direction. A recent structural model proposes that the C-ring has two distinct states (CCW and CW), in which torque helices in FliG_CC_ are located at different positions in the rotor–stator interface. Stator units interact with these two distinct states of the rotor. However, a detailed mechanism of how the conformational changes in FliG_M_ induced by CheY-P binding to FliM or CheY dissociating from FliM trigger the rearrangement of the torque helix into two distinct rotor states is still unclear.

To clarify the molecular mechanism underlying the determination of rotational direction in the motor, structural [18,28,29,40,41,42,43,44,45,46,47,48] and mutational [39,49,50,51,52,53,54] analyses, including molecular dynamics simulations, have been performed intensively on *Escherichia coli*, *Salmonella*, *Helicobacter pylori*, *Thermotoga maritima*, and *Borrelia* species. However, the variety of structures and the phenotypes of the mutations in the different species complicate finding a clear explanation of the detailed mechanism. We have performed mutational and structural analyses to clarify how conformational changes in FliG determine the rotational direction in marine *Vibrio* [1,7,8,55,56,57] (Appendix A). We found that the nonmotile PomB mutant can swim when combined with rotation-biased mutations in *the fliG* or *cheY* deletion strains, although the mechanism is still unclear [56]. In the present study, we tested whether the nonmotile FliG mutant can swim when combined with a rotation-locked mutation. We found that one nonmotile mutation in FliG (L259Q) was suppressed when expressed in the *fliG*/*cheY* deletion strain, although other mutations in FliG (L270P and L271R) were still nonmotile. The L259 residue is located on the C-terminal side of the MFXF motif. To understand how this suppression occurs, we introduced several mutations around the L259 residue in *Va* FliG and evaluated its motor functions. Using a characterization of L259Q and nearby mutations and a structural model of a *Vibrio* C-ring generated with cryo-electron tomography, we propose a model of the hydrophobic core’s role in FliG_C_ in determining the rotational direction of the motor.

## 2. Materials and Methods

### 2.1. Bacterial Strains, Media, and Growth Conditions

The bacterial strains and plasmids used in this study are listed in Table 1. *V. alginolyticus* cells were cultured at 30 °C in VC medium (0.5% [wt/vol] hipolypeptone, 0.5% [wt/vol] yeast extract, 0.4% [wt/vol] K_2_HPO_4_, 3% [wt/vol] NaCl, 0.2% [wt/vol] glucose) or in VPG medium (1% [wt/vol] hipolypeptone, 0.4% [wt/vol] K_2_HPO_4_, 3% [wt/vol] NaCl, 0.5% [wt/vol] glycerol). If needed, chloramphenicol was added to a final concentration of 2.5 µg/mL. *E. coli* cells were cultured at 37 °C in LB medium (1% [wt/vol] bactotryptone, 0.5% [wt/vol] yeast extract, 0.5% [wt/vol] NaCl). If needed, chloramphenicol was added at a final concentration of 25 µg/mL.

### 2.2. Mutagenesis

To introduce mutations into the *fliG* gene in the pNT1 plasmid, site-directed mutagenesis was carried out using the QuikChange method, as described by the manufacturer (Stratagene, La Jolla, CA, USA). The nucleotide sequences of the primers are listed in Appendix A. All constructs were confirmed via DNA sequencing. The transformation of *V*. *alginolyticus* with the pNT1 plasmid was performed through conjugational transfer from *E. coli* S17-1, as previously described [58]. Transformation of *E. coli* with the pNT1plasmid was performed using a conventional method.

### 2.3. Motility Assay in a Soft-Agar Plate

The cells with a Δ*fliG* mutation (NMB198) harboring the plasmid encoding FliG with or without mutations (pNT1) were grown overnight in VC medium at 30 °C. Overnight cultures (2 µL) were spotted in a VPG soft-agar plate containing 0.25% (*w*/*v*) bactoagar and 1 mM of isopropyl β-D-1-thiogalactopyranoside (IPTG) to induce FliG in pNT1. The soft-agar plates were then incubated for 6.5 h at 30 °C and photographed. Cells of the same strain harboring an empty plasmid (pMMB206) or a plasmid encoding wild-type FliG were used as the negative and positive controls, respectively.

### 2.4. Measurement of Motility Fraction and Swimming Speed by Dark-Field Microscopy

Cells were grown overnight in VC medium including chloramphenicol at 30 °C. They were diluted 100-fold in VPG medium with chloramphenicol and cultured for 4 h with shaking. The suspension was diluted 100-fold with V buffer (50 mM Tris-HCl [pH 7.5], 5 mM MgCl_2_, and 300 mM NaCl). Diluted suspension (2 µL) was then spotted on a slide glass for observation using dark-field microscopy and video recording.

To analyze the motility fraction, the ratio of swimming bacterial cells was calculated from all bacterial cells in the video. For each mutant, at least 100 cells were counted per sample, and the experiments were repeated three times.

To measure swimming speed, L-serine was added at a final concentration of 5 mM before spotting on the glass slide. The distance of the swimming cell for 1 s was tracked using ImageJ software (1.54 f). For each mutant, at least 20 cells were tracked in each video sequence.

### 2.5. Analysis of Rotational Direction and Switching Frequency

Cells were grown overnight in VC medium at 30 °C, diluted 100-fold in VPG medium, and cultured for 4 h with shaking. The suspension was then washed twice with V buffer. Six microliters of the suspension were spotted on a glass slide and covered with a coverslip for observation using high-intensity dark-field microscopy. Swimming cells were tracked for at least 10 s, and video images were recorded. We analyzed the rotational direction of the flagellum and counted the switching frequency over 10 s. The flagellar rotational direction was determined based on the position of the polar flagellum (front or back of each swimming cell) and the swimming direction of the cell. The flagellum pushes the cell body in a CCW direction and pulls it in a CW direction. At least 10 cells from each mutant were analyzed.

## 3. Results

### 3.1. Characterization of Motility Defect Mutations in FliG in the ΔfliGΔcheY Strain

We previously found several mutations in *Vibrio* FliG that confer defects in polar flagellar formation (Pof^−^), polar flagellar motility (Pom^−^), and chemotaxis (Che^−^) [56]. We also found that the Pom^−^ Che^−^ double mutant strain (Δ*pomAB*Δ*cheY*) can swim when nonmotile PomA/PomB (L160C/I186C) is expressed. PomB (L160C/I186C) forms an intramolecular disulfide bridge between two introduced Cys mutations in the PomB PGB domain that impair motility, presumably because the disulfide bridge inhibits the conformational changes required for interaction with the rotor [56]. This evidence suggests that fixing the C-ring conformation due to the Che^−^ mutation somehow suppresses the Mot^−^ (and/or Fla^−^) phenotype to restore the interaction between the stator and rotor. To extend this idea to nonmotile FliG mutants, we focused on the FliG-L259Q, L270P, and L271P mutations (Figure 1B), which confer the Mot^−^ phenotype [55]. Wild-type and mutant FliG variants were expressed in the Δ*fliG* (NMB198) and Δ*fliG*Δ*cheY* strains (NMB318) using a plasmid (pNT1, hereafter p*fliG*), and motility was observed under dark-field microscopy (Figure 1C). The motility fraction of the cells expressing wild-type (WT)-FliG was 0.31 in the Δ*fliG* and Δ*fliG*Δ*cheY* cells, whereas those expressing the FliG-L270P and L271R mutations were completely nonmotile. Δ*fliG* cells expressing FliG-L259Q were nonmotile, but few of the Δ*fliG*Δ*cheY* cells swam (fraction of 0.06), indicating that the motility defect due to the FliG-L259Q mutation was suppressed when combined with the Che^−^ mutation.

### 3.2. Swimming Fraction and Speed of the FliG-G214S/L259Q and G215A/L259Q Mutants

The rotational direction of the motor in the Δ*cheY* strain was fixed in the counterclockwise (CCW) direction. Our question is whether rotational fixation in the clockwise (CW) direction suppresses the motility defect caused by the FliG-L259Q mutation. We previously identified two *FliG* mutations, G214S and G215A, that confer CCW-biased and CW-locked rotations, respectively [56]. We generated the double mutants FliG-G214S/L259Q and G215A/L259Q, expressed them from p*fliG* in the Δ*fliG* and Δ*fliG*Δ*cheY* strains, and observed their motility under dark-field microscopy (Figure 2A). The motility fractions of the cells with the G214S and G215A mutants were 0.17 and 0.13 in the Δ*fliG* cells and 0.14 and 0.11 in the Δ*fliG*Δ*cheY* cells, respectively. We found that 0.06 of the cells expressing the G214S/L259Q mutant in both the Δ*fliG* and Δ*fliG*Δ*cheY* strains displayed swimming behavior, whereas the cells expressing the G215A/L259Q mutant in both the Δ*fliG* and Δ*fliG*Δ*cheY* strains did not swim at all. These results indicate that the motility defect in the FliG-L259Q mutation was intragenically suppressed in the CCW-biased mutation (G214S) but not in the CW-biased mutation.

Next, we measured the swimming speed of the Δ*fliG* and Δ*fliG*Δ*cheY* cells with the WT-FliG, G214S, L259Q, and G214S/L259Q mutants (Figure 2B). The swimming speeds of the cells with WT-FliG were 29 and 32 µm/s in the Δ*fliG* and Δ*fliG*Δ*cheY* strains, respectively. Cells expressing the FliG-G214S mutant swam at similar speeds to that of the WT-FliG in both the Δ*fliG* and Δ*fliG*Δ*cheY* strains. On the other hand, the Δ*fliG*Δ*cheY* cells with the L259Q mutation swam at 5.6 µm/s, five times slower than that of the WT and FliG-G214S mutant. Furthermore, Δ*fliG* and Δ*fliG*Δ*cheY* cells with the G214S/L259Q double mutation swam at similar speeds to those with the L259Q mutant.

### 3.3. Mutational Analysis of the Core of FliG_C_

In the homology model of *Vibrio* FliG [59], based on the FliG from *A. aeolicus* [27], the L259 residue is located near the MFXF-motif (M253, F254, V255, and F256 residues) of FliG_C_. This residue participates in forming the hydrophobic core of FliG_C_ through a loop–helix interaction (FliG_C_ α8; Figure 3A), which is involved in the F254, I328, L329, and A332 residues. We hypothesized that this hydrophobic core formation contributes to the flexibility of the MFXF motif and that inserting the hydrophobic mutation (L259Q) destabilizes this flexibility. We also expect that the flexibility of the MFXF motif affects the determination of the motors’ rotational direction; thus, we introduced various mutations into nearby residues of the MFXF motif (D251R, M253I, V255G, V255W, F256A, and E257K) and the α8 helix (Q325F, L329Q, A332F, and R333D) (Figure 3A) to test this idea. Charge inversion mutations D251R, E257K, and R333D were used to change the interaction surface between FliG_CN_ and FliG_CC_. To form additional hydrophobic interactions with the phenylamine ring of the F254 and F256 residues and affect the flexibility of the MFXF motif, we introduced the Q325F mutation. As the L329 residue is close to the L259 residue, we introduced the L329Q mutation to obtain a phenotype similar to that of the L259Q mutation. Other mutations were used to change the volume of the side chain from small to bulky, or vice versa, to alter the hydrophobic interactions around the L259 residue.

First, swimming motility assays were performed on soft-agar plates (Figure 3B). Wild-type or mutant FliG was expressed from the pNT1 plasmid in the Δ*fliG* strain, and if cells were motile, they formed a motility ring after incubation for 6.5 h. The motility rings of strains expressing V255G, V255W, and A332F mutations were similar in size to those of the WT, whereas those expressing M253I, F256A, E257K, Q325F, L329Q, and R333D mutations were smaller than those of the WT control. The D251R mutant did not form a swimming ring, indicating that this mutation resulted in the Fla^−^, Mot^−^, or Che^−^ phenotype.

Next, we measured the motility fraction and swimming speeds of cells expressing the FliG mutants (Figure 3C,D). The motility of the cells with V255G and A332F mutations was similar to that of the WT-FliG cells, whereas those with the D251R, M253I, V255W, F256A, E257K, L329Q, Q325F, and R333D mutations had motility that was reduced by half or less. In contrast to the motility fraction, the swimming speeds of cells with the mutations were approximately 28~42 µm/s. Thus, no *fliG* mutations showed the Mot^−^ phenotype, contrary to our expectations.

Finally, we measured the switching frequency (Figure 4A) and duration of the rotational direction (Figure 4B). The switching frequency was defined as the number of times each cell switched swimming direction within 10 sec (Figure 4A). In cells expressing WT-FliG, the motor switched rotational direction 13 times per 10 sec, and the ratio of CCW to CW was 0.70 to 0.30, consistent with our previous report [56]. The switching frequency of the motor with the six FliG mutants (V255G, F256A, E257K, Q325F, L329Q, and A332F) did not change significantly compared with that of the WT (*p* > 0.05; Student’s *t*-test). The motor with V255G and A332F mutations showed a similar CCW-CW ratio to the WT, whereas the motors with D251R and L329Q mutations conferred strong CW-biased rotation, showing CCW-CW ratios of 0.01 to 0.99 and 0.7 to 0.93, respectively. The motors with F256A, E257K, and Q325F mutations conferred CW-biased rotation, but they were not as strong as D251R and L329Q, with CCW-CW ratios of 0.44 to 0.56, 0.21 to 0.79, and 0.26 to 0.34, respectively.

## 4. Discussion

We found that the FliG-L259Q mutation, which results in defective motility (Mot^−^) [55], was genetically suppressed by the mutation of *cheY*. Furthermore, we found that cells with the FliG-G214S/L259Q mutation are motile, although cells with the FliG-G215A/L259Q mutation are nonmotile, indicating that the CCW-biased mutation (G214S and Δ*cheY*) suppressed the motility defect caused by the L259Q mutation but not the CW-biased rotation.

The L259 residue is located close to the MFXF motif of FliG_C_ and participates in hydrophobic interaction with the α8 of FliG_C_ (Figure 3A). Structural analysis was performed by comparing the solution NMR for the FliG_MC_ fragment with the L259Q mutation, revealing a unistructural pattern in the spectra [63]; however, cells form flagella normally [55]. Together with our results, this suggests that FliG with the L259Q mutation can form a C-ring complex, but this abolishes motor rotation, probably due to the disruption of the hydrophobic interaction and/or the disruption of flexibility in the MFXF motif. Since the *cheY* deletion and the FliG-G214S mutation cause the conformational transition of FliG into the CCW state, conformational rearrangement also occurred in FliG_CN_ and FliG_CN_ with the L259Q mutation. Therefore, we hypothesized that the motility defect caused by the L259Q mutation is suppressed by the CCW conformation of the C-ring. To elucidate the effect of structural disruption caused by the mutation on motor rotation, we introduced ten mutations into FliG at the neighboring residues of the MFXF motif and measured motility and motor functions. We found that the FliG-D251R and L329Q mutations conferred CW-biased rotation. These mutations are located on the N-terminal side of the MFXF motif and in the α8 of FliG_C_, indicating that disrupting the hydrophobic core in residues near the MFXF motif and the α8 of FliG_C_ affects the rotational direction. Notably, we did not find a Mot^−^ mutation similar to the L259Q mutation.

Together with the previous analysis of the rotational regulation of *Vibrio* FliG, the positions of the FliG mutations are mapped in Figure 5A,B, showing atomic models of FliG in the CCW and CW states, respectively, from structural analysis using cryo-electron tomography [21]. We previously analyzed the conformational changes of the G215A mutation using solution NMR and molecular dynamics (MD) simulations [57,59], which suggested that rearranging the hydrophobic interaction between ARM_C_ and the MFXF motif regulates the orientation of the torque helix, although the transition mechanism of the conformational change in FliG_C_ was unclear because we only found that the A282T mutation conferred the CW-biased rotation. The D251R, L259Q, and L329Q mutations from the analysis in this study contribute to our further understanding of the mechanism of rotational determination due to conformational changes in FliG; the formation of a hydrophobic core caused by C-terminal residues next to the MFXF motif determines the orientation of the torque helix. A previous study using NMR and MD simulations of the A282T mutation suggested that the A282T mutation forms an additional hydrogen bond with the V279 residue [63], located in the same helix as the A282 residue, in the CW state but not in the CCW state. This bond blocks the conformational transition of FliG_C_ from the CW to the CCW state. The L329 residue also forms a hydrophobic core in FliG_C_; thus, the L329Q mutation may affect the orientation of the torque helix. Given that the D251 residue is located on the N-terminal side of the MFXF motif, the D251R mutation probably disrupts the flexibility of the MFXF motif.

Taken together with the rotationally biased mutations, we suggest that rotational determination via FliG involves the following three steps (Figure 5A–C): (1) receiving a chemotactic signal at FliG_M_ due to CheY-P binding to FliM/FliN or CheY dissociating from FliM/FliN in the C-ring structure; (2) the impact on FliG_M_ affects the conformational dynamics of FliG_C_, including the EHPQR motif, the Gly-Gly flexible linker, and the MFXF motif; and (3) this induces the rearrangement of the hydrophobic interaction in the FliG_C_ core, thereby determining the orientation of the torque helix. The idea that the orientation of the torque helix determines the rotational direction is consistent with a previously proposed model stating that the rotational direction of the motor is determined by changing the interaction surface between FliG and the pentameric ring of the A subunit of the stator based on a structural analysis of the C-ring and stator complex in several bacteria [18,49,64].

Finally, we propose a model to explain why the motility defect in the L259Q mutation was suppressed by mutations conferring a CCW-biased rotation (Figure 5D,E). In the CCW state of the C-ring, the side chains of F256, L259, V262, L270, I328, and L329 formed a hydrophobic core on the C-terminal side of the MFXF motif. We assumed that the L259Q mutation disrupts the core, although the second CCW mutation in FliG_C_ restores the hydrophobic core. In contrast to the CCW state, residues L259, I267, and V294 formed a hydrophobic core in the CW state, suggesting that the hydrophobic interaction was disrupted because only three residues (L259, I267, and V294) interacted with each other. Furthermore, the disruption of core formation by the L259Q mutation may restrict the flexibility of the MFXF motif. Therefore, the mutation in the CW state did not suppress the motility defects caused by the L259Q mutation. The Mot^−^ phenotype caused by the L270P and L271R mutations was not rescued by the *cheY* deletion. We previously analyzed the conformation of the FliG_C_ fragment by performing NMR measurements [63], which indicated that the FliG_C_ domain was disordered due to the disruption of the hydrophobic core. Therefore, the CW state of FliG did not rescue its functional conformation.

## 5. Conclusions

In this study, we focused on the suppression of the motility defect caused by the L259Q mutation of *Vibrio* FliG in the Δ*fliG*Δ*cheY* strain. We showed that the suppression only occurred in the CCW rotation state, probably due to the rearrangement of the disrupted hydrophobic core formation. Furthermore, we characterized the FliG mutations neighboring the MFXF motif, suggesting that hydrophobic core formation and MFXF motif flexibility contribute to determining the rotational direction. The C-ring structures in the CCW and CW states are available in many species, although an explanation of the conformational transitions between these states is unclear. The proposed model can help us understand the conformational transitions of monomeric FliG. Filling the gap regarding the conformational transition and cooperativity between the monomeric and multimeric states will be our future focus.

## Figures and Tables

**Figure 1 biomolecules-15-00212-f001:**
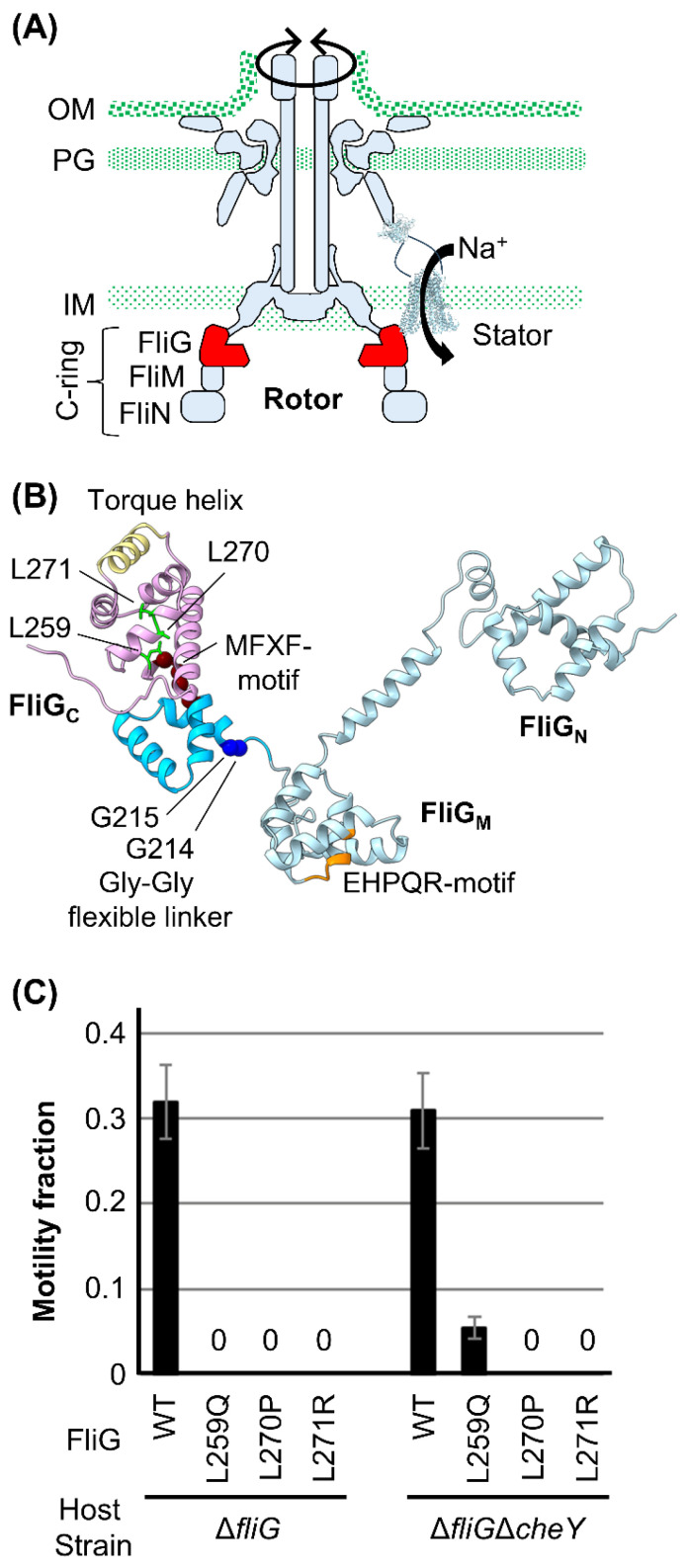
Flagellar motor and FliG in marine *Vibrio*. (**A**) Schematic model of the sodium-driven polar flagellar motor in *Vibrio*. A sodium ion channel unit called a stator couples with the sodium ion flux to drive the rotor by interacting with the C-ring protein FliG. The C-ring comprises FliG, FliM, and FliN. IM, inner membrane; PG, peptidoglycan layer; OM, outer membrane. (**B**) Homology model of *Va* FliG based on the crystal structure of *A*. *aeolicus* FliG (PDB code, 3HJL) [27]. FliG comprises three domains: the N-terminal (FliG_N_), middle (FliG_M_), and C-terminal (FliG_C_). FliG_N_ and FliG_M_ are indicated by light blue in the ribbon model. The EHPQR motif (the FliM interaction site) is indicated by orange in the ribbon model. The Gly-Gly flexible linker region is located between FliG_M_ and FliG_C_ and is indicated by blue in the sphere model. The MFXF motif is brown in the sphere model and separates FliG_C_ from FliG_CN_ and FliG_CC_, shown in cyan and pink in the ribbon model, respectively. The interaction helix with the stator in FliG_CC_, the torque helix, is indicated by khaki in the ribbon model. The mutation sites at L259, L270, and L271 in FliG_CC_ are indicated by lime in the stick model. Based on the sequence alignment, corresponding residues of *Vibrio* FliG are indicated. (**C**) The motility fractions of the Δ*fliG* (NMB198) and Δ*fliG*Δ*cheY* (NMB318) strains harboring the pNT1 plasmid (encoding the wild-type or mutant FliG variant) were measured. A “0” means nonmotile. All experiments were repeated three times, and average values with standard deviations are shown.

**Figure 2 biomolecules-15-00212-f002:**
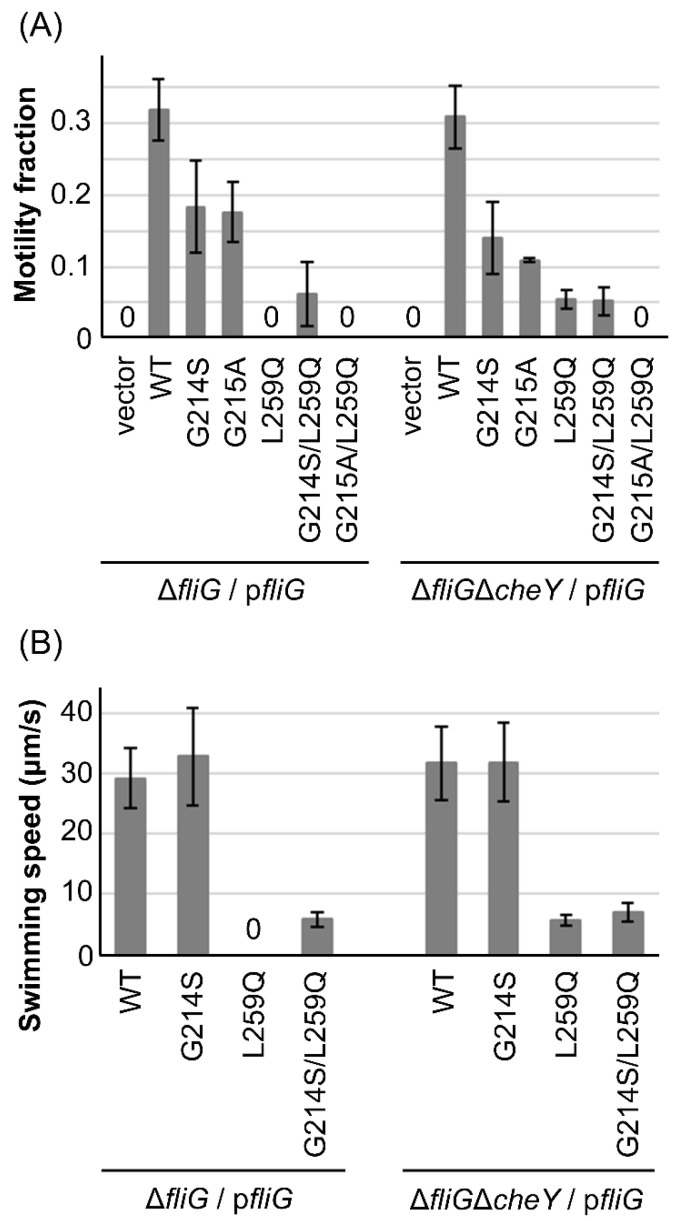
Characterization of the FliG-L259Q mutant. (**A**) The motility fractions of the Δ*fliG* and Δ*fliG*Δ*cheY* strains harboring pMMB206 (vector control) or pNT1 (shown here as p*fliG*)-encoding wild-type or *FliG* mutant derivatives were measured. All experiments were repeated three times, and average values with standard deviations are shown. A “0” indicates nonmotile. Data for the WT-FliG and L259Q mutation are the same as those shown in Figure 1C. (**B**) The swimming speeds of the Δ*fliG* and Δ*fliG*Δ*cheY* strains harboring pNT1 (shown here as p*fliG*)-encoding wild-type or FliG mutant derivatives were measured. The swimming speeds of 20 cells were measured with 5 mM L-serine as an attractant to induce smooth swimming, and average values with standard deviations are shown. A “0” indicates nonmotile.

**Figure 3 biomolecules-15-00212-f003:**
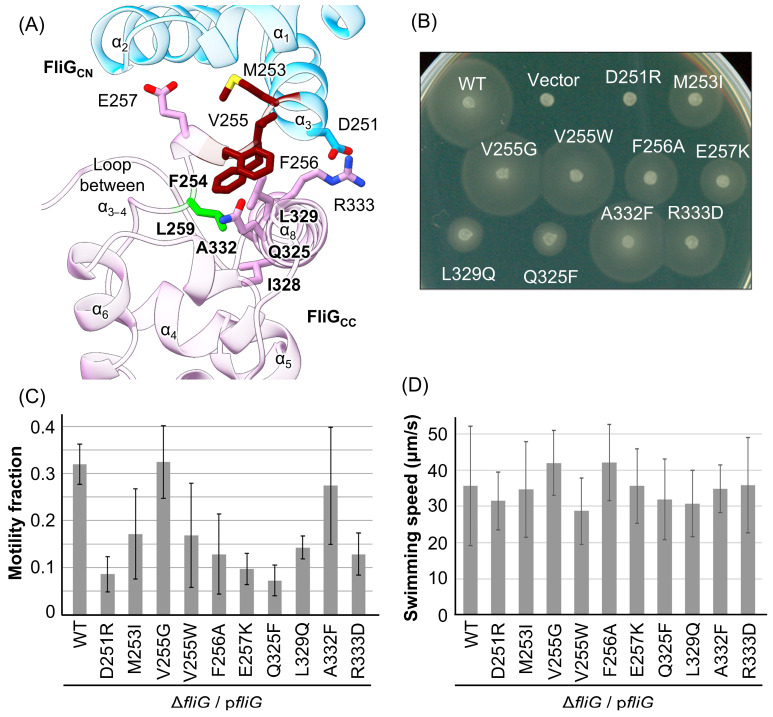
Effect of point mutations in the MFXF motif on motor function. (**A**) Homology model of *Vibrio* FliG_C_ based on the crystal structure of full-length FliG in *Aquifex aeolicus* (PDB code, 3HJL) [27]. FliG_C_ consists of eight α-helices (α1–8); the L259 residue and the MFXF motif are located at the loop between α3 and 4. FliG_CN_ and FliG_CC_ are indicated by cyan and pink, respectively, in the ribbon model. The MFXF motif is in brown in the ribbon and stick models. The side chains of the mutation sites on the D251, M253, V255, F256, E257, Q325, L329, A332, and R333 residues in *Va* are shown in the stick model. Nitrogen and oxygen atoms are blue and red, respectively. The L259 residue and its interaction partners are highlighted in bold text. (**B**) Motility assay on soft-agar plates: 2 µL of overnight cultures of NMB198 (Δ*fliG*) harboring the empty plasmid (pMMB206) or pNT1 (FliG, shown here as p*fliG*) with or without mutation was spotted on a soft-agar plate containing 0.25% (*w*/*v*) bactoagar with 1 mM IPTG and then incubated at 30 °C for 6.5 h. (**C**) Motility fractions of the Δ*fliG* strain expressing wild-type FliG or FliG mutant variants were measured. All experiments were repeated three times, and average values with standard deviations are shown. (**D**) The swimming speeds of the Δ*fliG* strain expressing wild-type FliG or mutant FliG variants were measured. The swimming speeds of 20 cells were measured with 5 mM L-serine as an attractant to induce smooth swimming, and average values with standard deviations are shown.

**Figure 4 biomolecules-15-00212-f004:**
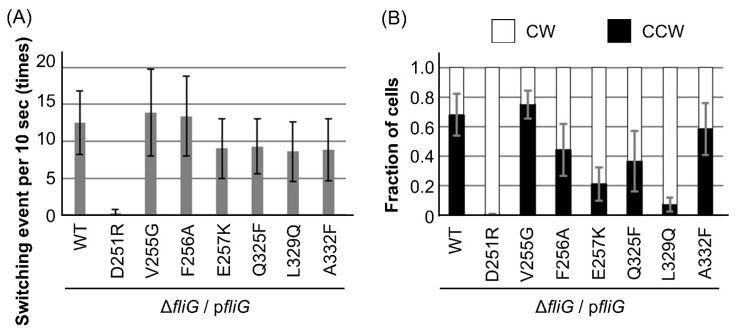
Characterization of motor rotation in the mutations. NMB198 (Δ*fliG*)-harboring pNT1 cells with or without mutations were grown and observed under high-intensity dark-field microscopy. (**A**) Switching events (total counts of switching from the CCW direction to the CW direction and vice versa) per 10 s. (**B**) The ratio of rotational durations in the counterclockwise (CCW) direction to the clockwise (CW) direction in the *fliG* mutations. All experiments in (**A**,**B**) were repeated at least ten times, and the average values with standard deviations are shown.

**Figure 5 biomolecules-15-00212-f005:**
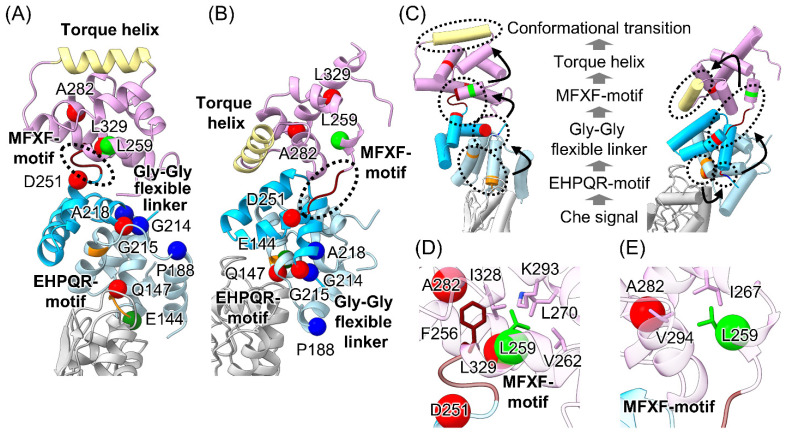
Map of the FliG mutations in counterclockwise (CCW) and clockwise (CW) states of the C-ring. (**A**,**B**) Ribbon model of CCW (**A**) and CW (**B**) states. The model was generated based on the C-ring volume by performing cryo-electron tomography analysis [21]. FliG_M_, FliG_CN_, FliG_CC_, and FliM are shown in light blue, cyan, pink, and gray, respectively. The green, yellow-green, blue, and red balls indicate the position of the mutations that resulted in increased motor switching (E144D), motility defects suppressed with the mutation of CCW rotation (L259Q), CCW-biased rotation (P188L, G214S, and A218V), and CW-biased rotation (Q147H, G215A, D251R, A282T, and L329Q), respectively. The EHPQR motif, MFXF motif (dotted line), and torque helix are orange, brown, and khaki, respectively. (**C**) Schematic model of conformational transition of FliG. The left and right panels show the CCW and CW states from (**A**) and (**B**), respectively, in the cylinder model. Dotted lines highlight the EHPQR motif, the Gly-Gly flexible linker, the MFXF motif, and the torque helix. This indicates the steps in the conformational transition of the EHPQR motif to the torque helix via the Gly-Gly flexible linker and the MFXF motif. (**D**,**E**) Structures around the L259 residue of the CCW (**D**) and CW (**E**) states from (**A**) and (**B**), respectively. The interaction partners with L259 are shown in the stick models.

**Table 1 biomolecules-15-00212-t001:** List of bacterial strains and plasmids.

Strains or Plasmids	Genotype or Description	Reference or Source
** *V. alginolyticus* **		
VIO5	Wild-type strain of a polar flagellum (Rif^+^ Pof^+^ Laf^−^)	[58]
NMB198	VIO5 *fliG* (Pof^−^)	[7]
NMB318	NMB198 Δ*cheY* (Pof^−^, Che^−^)	[59]
** *E. coli* **		
DH5α	Host for cloning experiments	[60]
S17-1	*recA hsdR thi pro ara RP-4 2-tc::Mu-Km::Tn7* (Tp^r^ Sm^r^)	[61]
BL21(DE3)	*F^−^*, *ompT*, *hsdS_B_*(r_B_^−^ m_B_^−^), *gal*(*λ_c_I* 857, *ind1*, *Sam7*, *nin5*, *lacUV*5-T7*gene*1), *dcm*(DE3) (Host for protein expression)	Novagen
**Plasmids**		
pMMB206	Cm^r^, P_tac_P_lac_UV5	[62]
pNT1	*fliG* in pMMB206	[8]

Rif^r^, rifampin resistant; Pof^+^, normal polar flagellum; Laf^−^, defective in lateral flagellar formation; Pof^−^, defective in polar flagellar formation; Che^−^, defective in chemotaxis; TP^r^, trimethoprim resistant; Sm^r^, streptomycin resistance; Cm^r^, chloramphenicol resistance; P_tac_, tac promotor; P_lac_, lac promotor.

## Data Availability

The data supporting this article have been included as part of the main figures or as Appendix A.

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
