# Peer review of "Regulatory Role of a Hydrophobic Core in the FliG C-Terminal Domain in the Rotary Direction of a Flagellar Motor"

_biomolecules, 2025, doi:10.3390/biom15020212_

Round 1
Reviewer 1 Report
Comments and Suggestions for Authors
The authors analyze find that a particular non-motile mutant in the flagellar rotor protein FliG of V. alginolyticus could be restored to motility by mutation of cheY. Moreover they find that any mutation that increases the stability of the CCW conformation of FliG will also suppress the mutation. The work is well done and the results are nice. I couldn’t quite grasp why they authors are invoking a regulatory role (e.g. title) for this residue. Couldn’t the CCW conformation simply physically restore the interaction? Also, if FliGL259Q motility could be restored by CCW rotation of the motor, why wouldn’t it present as motile in the swim plate assay? I mean, the gradient generated by local consumption and resulting chemotactic response, I presume, would bias the motor in the CCW direction as well. Even in homogeneous media, I would expect one should see some swimming in the FliGL259Q mutant but only when force is being generated in the CCW direction.
The only organizational complaint about the manuscript I have is that the text tends to bog down in speculative structural details and this makes it very difficult to understand exactly what the authors are concluding in terms of the overall biology of flagellar function and why they are concluding that.
A simple statement summarizing the main idea would be very beneficial. For example, the end of the abstract and discussion end on points I don’t quite understand.
Specific comments:
Does the single polar flagellum of V. alginolyticus generate force in both directions or only one?
Line 64. “On the other hand”. Clarify. On the other hand of what?
Para starting line 64. Paragraph is very long. Consider breaking into two or three paragraphs with separate points being made in each.
Line 126. I believe the “fliG/cheY deletion strain” should be just “cheY deletion strain”. The description of suppression already indicates the presence of the fliG, and the current text reads like both fliG and cheY are deleted.
Line 127. Paragraph ends abruptly without a point being made.
Table 1. NMB318 is described as a cheY mutant that is also polar flagella defective (pof-). Is that accurate? What causes the loss of polar flagella?
Line 187. “fla-“ Should this be “pof-“ for Vibrio? They should be the same thing, but pof- would keep the nomenclature consistent. Same for “mot-“ should be “pom-“?
Line 193. Doesn’t the interpretation that the conformation of FliG being able to restore motility to the motB disulfide knock in contradict the interpretation that the disulfide prevents MotB anchoring to the PG (re: PGB domain)? Or by “anchoring to the motor” do you mean “interaction with the rotor”? Clarify.
Line 195. I presume that the mot- interpretation of the mutant from reference 50 was based on the observation that the mutant was non-motile but still produced a flagellum (by EM?). Is this the case? Also, how was it shown to be non-motile? In swim plate assay? Some specifics here would be important given how special this allele is and the fact that the authors couldn’t phenocopy it when other mutations in the protein.
Line 218. Consider adding the word “intragenically” before “suppressed” to distinguish this result from the extragenic suppression by deletion of cheY.
Line 314. Should “suppresses swimming due to the cheY deletion” be substituted with “was genetically suppressed by mutation of cheY”?
Para starting 337. Parragraph is very long and the point gets lost. Consider simplifying and breaking into multiple paragraphs if necessary.
Reviewer 2 Report
Comments and Suggestions for Authors
General comments
1. Abstract is dense clearly outline research gap.
2. Introduction provides enough background on the flagellar motor and FliG, however research gap and objectives are not clear.
3. why selected L259Q for mutation?
4. Results presented in excessive detail focus on key findings avoid redundancy
5. Minor language editing is needed.
6. Figures 1and 2 are informative but visually crowded.
Specific Comments
Introduction
Line-35-42: link the description of flagellum to the research question
Line-50-55: stator and rotor interactions are dense simplify
Line-95-98: How this study is novel? Which question need to be solved here?
M & M
Line 146-151: specify primer design and mutagenesis efficiency
Discussion
Line 348-362: Need a clearer visual representation
Figures-1-4: need full explanation and their significance in captions.
Comments on the Quality of English Language
Minor language editing is needed.
Author Response
Reviewer2:
Comments
The manuscript entitled “The regulatory role of the hydrophobic core in the FliG Cterminal domain on the rotary direction of the bacterial flagellar motor” is an interesting topic, however, novelty with existing research is important. I would recommend to accept the paper.
Response: Thank you very much for the review and favorable comment to this manuscript.
General comments
- Abstract is dense clearly outline research gap.
Response: We tried to rewrite the abstract according to the comment.
- Introduction provides enough background on the flagellar motor and FliG, however research gap and objectives are not clear.
Response: We added some explanations in the introduction.
- why selected L259Q for mutation?
Response: Three FliG mutations (L259Q, L270P, and L271R) showed nonmotile reported in ref 56, and the L259Q mutation was only suppressed in the cheY/fliG strain. We added the sentence at lane 131, ” although other mutations in FliG (L270P and L271R) were still nonmotile”.
- Results presented in excessive detail focus on key findings avoid redundancy
Response: We deleted the sentences “We previously performed structural analysis of the L259Q mutation in the Vibrio FliG fragment, which contained FliGM and FliGC (FliGMC), by measuring the spectra of nuclear magnetic resonance (NMR) in solution state, and found that the spectrum pattern of the FliGMC fragment with the L259Q mutation was unstructured [57].” and ” Vibrio alginolyticus has a single polar flagellum and swims forward in liquid by rotating the flagellum in the CCW direction and backward by rotating it in the CW direction. Therefore, we measured the duration of each rotational direction by observing the swimming cells under high-intensity dark-field microscopy (Fig. 4B).”
- Minor language editing is needed.
Response: We utilized English language editing service by MDPI.
- Figures 1and 2 are informative but visually crowded.
Response: We changed font sizes and several positions of the caption in Figure 1B. In Figure 1C, 2A and 2B, blank space is inserted between fliG and fliG/cheY strains and font sizes were changed.
Specific Comments
Introduction
Line-35-42: link the description of flagellum to the research question
Response: We added a sentence the end of the first paragraph.
Line-50-55: stator and rotor interactions are dense simplify
Response: Characterization of the stator unit is key for understanding the molecular mechanism of the torque generation. Thus, we think these sentences are essential, however, we tried to shorten the text.
Line-95-98: How this study is novel? Which question need to be solved here?
Response: We added the sentences in the last paragraph to answer them, as also another reviewer commented.
M & M
Line 146-151: specify primer design and mutagenesis efficiency
Response: We added Table S2 and showed the primer sequences.
Discussion
Line 348-362: Need a clearer visual representation
Response: We inserted new figure between Figure 5B and C to explain the steps of the conformational transition of FliG.
Figures-1-4: need full explanation and their significance in captions.
Response: We added and corrected the sentences in each caption.